# Design and experimental validation of multi-section directional coupler with arbitrary coupling and high directivity for sub-6 GHz UWB applications

Shrawan K. Patel[1], Rusan Kumar Barik[2*], Niraj Kumar Dewangan[3*], Slawomir Koziel[4,5]

1 Department of Electronics and Communication Engineering, School of Engineering and Technology, GGV Bilaspur, Chhattisgarh, India, 2 Department of Electronics and Communication Engineering, School of Engineering and Technology, CHRIST University, Bangalore, India, 3 Manipal Institute of Technology, Manipal Academy of Higher Education, Manipal, India, 4 Engineering Optimization and Modeling Center, Reykjavik University, Reykjavik, Iceland, 5 Faculty of Electronics, Telecommunications and Informatics, Gdansk University of Technology, Gdansk, Poland

* niraj.dewangan@manipal.edu (NKD), rusankumar.barik@christuniversity.in (RKB)

## Abstract

This work presents a geometrically simple topology for developing an ultra-wideband directional coupler with improved coupling and directivity. A short-ended coupled-line structure is used to achieve an ultra-wideband, tightly coupled symmetric three-section coupler using the microstrip line technology. The proposed design demonstrates an explicit improvement of approximately 1.2 dB in coupling compared to conventional multi-section directional couplers. Calculated, simulated, and measured responses validate the effectiveness of the proposed configuration in terms of low-ripple coupling bandwidth, low insertion loss, and improved directivity performance compared to respective responses of the conventional structure. Couplers featuring a higher number of sections to implement different bandwidths and couplings can be fabricated using the presented structure due to its transmission line-based approach. A prototype of the three-section directional coupler with coupling of 7.6 dB, 8.1 dB, and 8.3 dB and corresponding bandwidths of 104%, 123% and 133% is designed, fabricated, and measured. The experimental results confirm that the coupler can reliably achieve higher coupling with ultra-wideband response from 0.75 GHz to 3.75 GHz (5:1) with $8.3 \pm 1.4$ dB (ripple). Additionally, the design yields promising performance with return loss > 16 dB, isolation > 20 dB, a phase difference of $90 \pm 4°$, and directivity > 30 dB, and the maximum circuit size is $0.067\lambda_0^2$. This work aligns with SDG 9: Industry, Innovation and Infrastructure by advancing high-performance microwave components that support efficient, reliable, and scalable communication infrastructure.

which permits unrestricted use, distribution, and reproduction in any medium, provided the original author and source are credited.

**Data availability statement:** All relevant data are within the paper and its Supporting Information files.

**Funding:** The author(s) received no specific funding for this work.

**Competing interests:** The authors declare that there are no conflicts of interest.

## 1. Introduction

Directional couplers employing parallel microstrip coupled transmission lines [1,2] are extensively used in a range of radio frequency (RF) and microwave applications, such as Nolen matrix-based RF beamforming network [3], phase shifting, power combining /dividing, and power sampling, etc. [4] due to their convenient integration and compatibility with other circuitry. Weaker-coupling directional couplers are frequently employed in reflectometers, power-level meters, automatic gain control loops, and similar measurement applications [5–7]. There is a preference for directional couplers that can operate over a wide frequency range, enabling measurements without the limitations imposed by the measured frequency. Among the various types of directional couplers, multi-section coupled-line directional couplers are known for their broadband performance. Their bandwidth is related to the number of sections utilized and the selected coupling response ripple.

Earlier, two distinct categories of multi-section coupled-line couplers have been documented: symmetrical coupled multi-section coupled-line couplers, which maintain a consistent 90° phase difference between the coupled waves in the coupler region regardless of the signal frequency [8,9]. Asymmetric coupled-line couplers, which are approximately half the size of their symmetrical counterparts, have been favored over due to their smaller size [10,11]. However, in applications such as diplexers, multiplexers, directional filters, balanced mixers, and other devices where phase relationship is crucial and even-odd mode mathematical analysis is of prime relevance, symmetrical structures are preferred. Asymmetrical couplers lack this phase property and are primarily utilized for broadband power division without requiring precise phase control.

Achieving tight coupling in microstrip directional couplers is challenging. This challenge can be addressed by employing Lange couplers [12]. Nevertheless, Lange couplers require high-permittivity substrates to achieve tight coupling and are unsuitable for interconnections within multi-section couplers due to their structural complexity. A notable structure is the slot-coupled directional coupler [13], which can achieve tight coupling. However, its primary drawback is the limited bandwidth. Tight coupling has been achieved using re-entrant coupled lines reported in [14], which is again a multilayer topology. Realization as a tandem connection of two weakly coupled sections for tight coupling in [15]. However, the downside of tandem couplers is their larger physical size, as their overall electrical length is twice that of conventional directional couplers. Vertically installed planar couplers have been employed in the tight-coupling sections [16], yielding favorable outcomes. Nevertheless, this design is more difficult to manufacture compared to purely planar structures. To enhance the bandwidth of the coupler structure for ultra-wideband applications, various solutions have been proposed, such as elliptical shaping implemented in [17,18]. However, due to the elliptical shape sections in multilayer designs, their development is complicated. Unequal lengths coupled line structure is used in [19], MMIC technology is used in [20], slot coupled multi-section in [21], non-uniform couplers technique in [22,23], parasitic compensation techniques in [24–27], defected ground structures in [28], dielectric overlay technique in [29,30] and slot-line using asymmetric parallel

loaded branches in [31]. However, attractive features such as a simple, single-layer planar structure and the symmetric structure of the classical directional coupler are lost, and the design procedure becomes more complex due to multilayer complexity.

In this work, the possibility of enhancing the coupling of traditional directional couplers, such as asymmetrical coupled line-based single-section directional coupler (SSDC-ACL) and symmetrical coupled line-based single-section directional coupler (SSDC-SCL) shown in Fig 1 and conventional multi-section directional coupler (C-MSDC) shown in Fig 2, by incorporating a small length coupled-line section without affecting the ultra-wideband performance and size of the structure, is presented, as the proposed multi-section directional coupler (P-MSDC), shown in Fig 3.

Table 1 lists the various even–odd mode impedance ratios $\rho$ and the electrical lengths of coupled microstrip lines $\theta$. A 3-section symmetric multi-section coupler for the bandwidth equal 3.13:1 (with the center frequency $f_0 = 2.4$ GHz), with a theoretical coupling of $10 \pm 0.2$ dB [32], has been chosen to achieve the frequency range from 0.75 to 3.75 GHz based on the fabrication limit. When contrasted with existing ultra-wideband and tightly coupled couplers, the proposed structure is simpler to design than broadside couplers, multilayer designs, and other ultra-wideband approaches. Results of the presented structure validate its benefits over traditional three-section directional couplers, including reduced size due to coupled lines in the first and third sections, a higher bandwidth ratio, improved loss performance, and flat directivity and

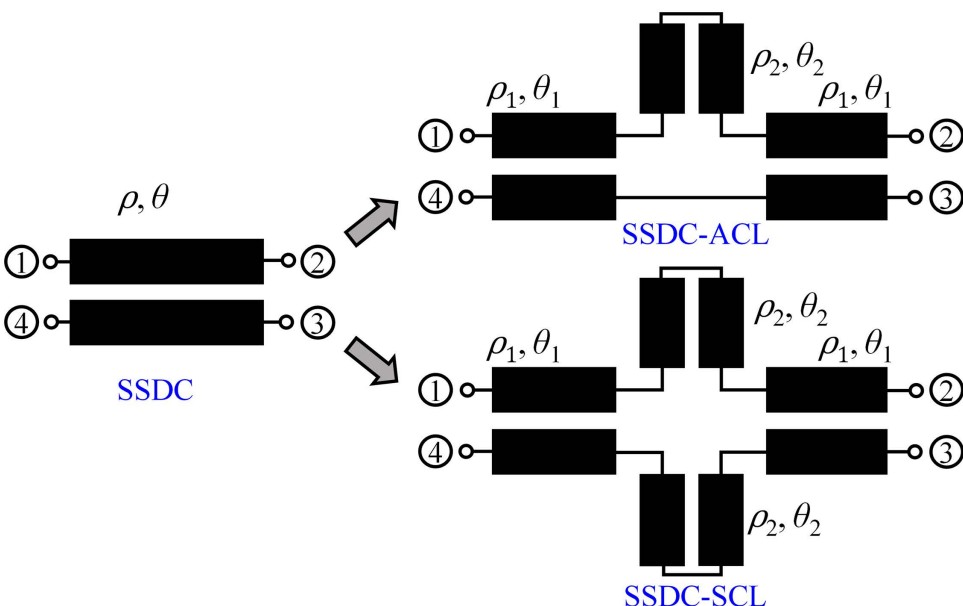

**Fig 1. Schematic of different topologies of single-section directional couplers.**

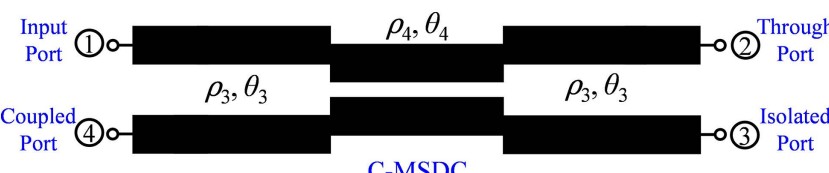

**Fig 2. Schematic of conventional multi-section directional coupler.**

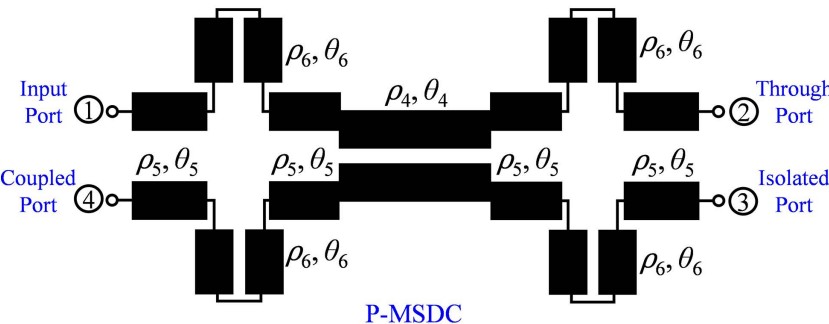

**Fig 3. Schematic of proposed multi-section directional coupler.**

**Table 1. Transmission line parameters of couplers of Fig 1 ($\theta$ in degrees).**

| $\rho=\rho_1=\rho_2=\rho_4=\rho_6$ | $\rho_3=\rho_5$ | $\theta=\theta_3=\theta_4$ | $\theta_1+\theta_2$ | $\theta_5+\theta_6$ |
|---|---|---|---|---|
| 2.48 | 1.14 | 90 | 180 (ACL) 45 (SCL) | 45 |

phase response. Due to additional coupled lines, perturbations in the first and third sections provide equalization of the even and odd modes.

Section II of this paper presents *S*-parameter analysis of the lossless transmission line-based P-MSDC and derives its matching and isolation conditions. Section III presents the design methodology from SSDC to P-MSDC along with its simulated results. Section IV shows the simulated and measured results of the prototype of P-MSDC.

## 2. Theoretical analysis of directional coupler unit

The overall even- and odd-mode ABCD matrices are attained by multiplying the ABCD matrices of each section evaluated under corresponding even- and odd-mode excitations [33–35]. Therefore, for a generalized multi-section directional coupler, the even and odd mode reflection and transmission coefficients in terms of even and odd mode elements of ABCD matrices are as follows:

$$\Gamma_{e,o} = \frac{A_{e,o} + \frac{B_{e,o}}{Z_0} - C_{e,o}Z_0 - D_{e,o}}{A_{e,o} + \frac{B_{e,o}}{Z_0} + C_{e,o}Z_0 + D_{e,o}},$$

(1a)

$$T_{e,o} = \frac{2}{A_{e,o} + \frac{B_{e,o}}{Z_0} + C_{e,o}Z_0 + D_{e,o}}.$$

(1b)

where $Z_0 = 50\ \Omega$ is the port reference impedance. Now, by using (1), the *S*-parameters of the directional coupler can be expressed as follows:

$$S_{11} = \frac{1}{2}\left(\Gamma_e + \Gamma_o\right), \ S_{21} = \frac{1}{2}\left(T_e + T_o\right),$$

$$S_{31} = \frac{1}{2}\left(T_e - T_o\right), \ S_{41} = \frac{1}{2}\left(\Gamma_e - \Gamma_o\right).$$

(2)

For any ideal coupled line directional coupler, which is completely reciprocal and lossless, $A_{e,o}$ and $D_{e,o}$ are pure real and $B_{e,o}$ and $C_{e,o}$ are pure imaginary [35]. Therefore,

$$A_e + D_e = A_o + D_o, \tag{3a}$$

$$\frac{B_e}{Z_0} + C_e Z_0 = \frac{B_o}{Z_0} + C_o Z_0. \tag{3b}$$

$$\left(\frac{B_e}{Z_0} - C_e Z_0\right) + \left(\frac{B_o}{Z_0} - C_o Z_0\right) = 0, \tag{3c}$$

$$and\ \frac{B_e}{Z_0} = C_o Z_0,\ \frac{B_o}{Z_0} = C_e Z_0. \tag{3d}$$

Applying these matching and isolation conditions, through and coupled S-parameter expressions can be calculated and shown their dependency primarily on $A_{e,o}$ and $B_{e,o}$ as

$$S_{21} = \frac{2Z_0}{2A_e Z_0 + (B_o + B_e)},\ S_{41} = \frac{(B_e - B_o)}{2A_e Z_0 + (B_e + B_o)} \tag{4}$$

## 2.1. Design and analysis of proposed multi-section directional coupler

Compared to C-MSDC shown in Fig 2, where three parallel coupled lines are connected in cascade, the P-MSDC topology also has three cascaded sections but the first and third sections have additional C-section coupled line structure, of similar even and odd mode impedances as the second section, are added. Schematic of even and odd mode half circuits of the presented configuration in terms of even and odd mode impedances and electrical length $\theta$, are shown in Figs 4, and Fig 5, respectively.

The elements of ABCD matrices for even and odd mode half circuits can be calculated by considering three sections in cascade, as shown in Figs 4, and Fig 5, as follows:

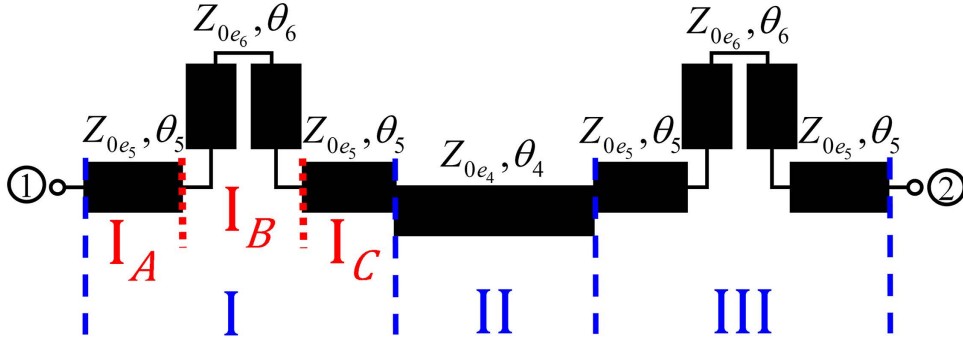

**Fig 4. Schematic of even mode excitations of the proposed multi-section directional coupler.**

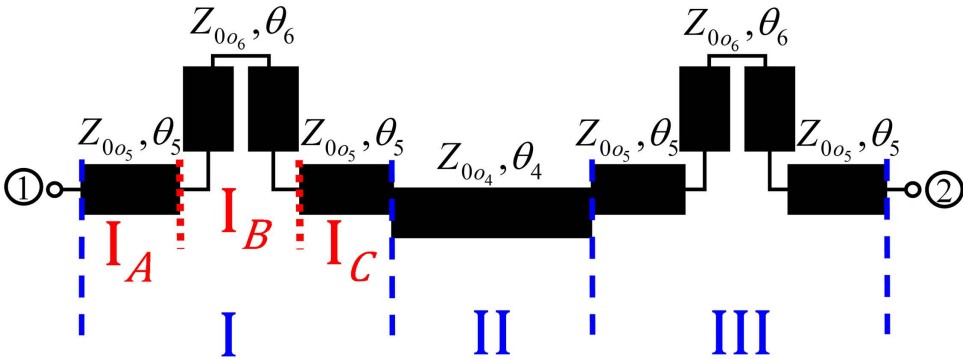

**Fig 5. Schematic of odd mode excitations of the proposed multi-section directional coupler.**

$$\begin{pmatrix} A_{e,o} & B_{e,o} \\ C_{e,o} & D_{e,o} \end{pmatrix} = \begin{pmatrix} A_{e,o_I} & B_{e,o_I} \\ C_{e,o_I} & D_{e,o_I} \end{pmatrix} \begin{pmatrix} A_{e,o_{II}} & B_{e,o_{II}} \\ C_{e,o_{II}} & D_{e,o_{II}} \end{pmatrix} \begin{pmatrix} A_{e,o_{III}} & B_{e,o_{III}} \\ C_{e,o_{III}} & D_{e,o_{III}} \end{pmatrix} \tag{5}$$

Here, $\begin{pmatrix} A_{e,o_I} & B_{e,o_I} \\ C_{e,o_I} & D_{e,o_I} \end{pmatrix} = \begin{pmatrix} A_{e,o_{III}} & B_{e,o_{III}} \\ C_{e,o_{III}} & D_{e,o_{III}} \end{pmatrix}$,

$$\begin{pmatrix} A_{e,o_I} & B_{e,o_I} \\ C_{e,o_I} & D_{e,o_I} \end{pmatrix} = \begin{pmatrix} A_{e,o_{IA}} & B_{e,o_{IA}} \\ C_{e,o_{IA}} & D_{e,o_{IA}} \end{pmatrix} \begin{pmatrix} A_{e,o_{IB}} & B_{e,o_{IB}} \\ C_{e,o_{IB}} & D_{e,o_{IB}} \end{pmatrix} \begin{pmatrix} A_{e,o_{IC}} & B_{e,o_{IC}} \\ C_{e,o_{IC}} & D_{e,o_{IC}} \end{pmatrix}$$

and $\begin{pmatrix} A_{e,o_{IA}} & B_{e,o_{IA}} \\ C_{e,o_{IA}} & D_{e,o_{IA}} \end{pmatrix} = \begin{pmatrix} A_{e,o_{IC}} & B_{e,o_{IC}} \\ C_{e,o_{IC}} & D_{e,o_{IC}} \end{pmatrix}$

Even and odd mode $[A, B, C, D]_{IA}$, $[A, B, C, D]_{IB}$ [34], and hence equivalent even and odd mode $[A, B, C, D]$ parameter matrices are

$$A_{e,o_{IA}} = A_{e,o_{IC}} = \cos\theta_5, B_{e,o_{IA}} = B_{e,o_{IC}} = jZ_{0e,o_5}\sin\theta_5, C_{e,o_{IA}} = C_{e,o_{IC}} = \frac{j}{Z_{0e,o_5}}\sin\theta_5, D_{e,o_{IA}} = D_{e,o_{IC}} = \cos\theta_5. \tag{5a}$$

$$A_{e,o_{IB}} = \frac{Z_{0e_6}\cot\theta_6 - Z_{0o_6}\tan\theta_6}{Z_{0e_6}\cot\theta_6 + Z_{0o_6}\tan\theta_6}, B_{e,o_{IB}} = \frac{2jZ_{0e_6}Z_{0o_6}}{Z_{0e_6}\cot\theta_6 + Z_{0o_6}\tan\theta_6}, C_{e,o_{IB}} = \frac{2j}{Z_{0e_6}\cot\theta_6 + Z_{0o_6}\tan\theta_6}, D_{e,o_{IB}} = A_{e,o_{IB}} \tag{5b}$$

$$A_{e,o} = \frac{1}{Z_{0e,o_5}Z_{1e,o}^2} \left[ \begin{matrix} \left\{ Z_{0e,o_5}Z_{2e,o}\cos2\theta_5 - (Z_{0e,o_5}^2 + Z_{3e,o})\sin2\theta_5 \right\} \begin{Bmatrix} \cos\theta_4 \left( Z_{0e,o_5}Z_{2e,o}\cos2\theta_5 - (Z_{0e,o_5}^2 + Z_{3e,o})\sin2\theta_5 \right) \\ -\frac{\sin\theta_4}{Z_{0e,o_4}} \left( Z_{0e,o_5}Z_{2e,o}\sin2\theta_5 + 2Z_{4e,o} \right) \\ -\frac{Z_{0e,o_4}\sin\theta_4}{Z_{0e,o_5}^2} \left( Z_{0e,o_5}Z_{2e,o}\sin2\theta_5 + 2Z_{5e,o} \right) \end{Bmatrix} \\ -\left\{ \frac{\cos\theta_4}{Z_{0e,o_5}} \left( Z_{0e,o_5}Z_{2e,o}\sin2\theta_5 + 2Z_{4e,o} \right) \left( Z_{0e,o_5}Z_{2e,o}\sin2\theta_5 + 2Z_{5e,o} \right) \right\} \end{matrix} \right] \tag{5c}$$

$$B_{e,o} = \frac{j}{Z_{0e,o_5}Z_{1e,o}^2} \left[ \begin{matrix} \left\{ 2\cos\theta_4 \left( Z_{0e,o_5}Z_{2e,o}\sin2\theta_5 + 2Z_{4e,o} \right) \left( Z_{0e,o_5}Z_{2e,o}\cos2\theta_5 - (Z_{0e,o_5}^2 + Z_{3e,o})\sin2\theta_5 \right) \right\} \\ + \left\{ Z_{0e,o_4}\sin\theta_4 \left( Z_{0e,o_5}Z_{2e,o}\cos2\theta_5 - (Z_{0e,o_5}^2 + Z_{3e,o})\sin2\theta_5 \right)^2 \right\} - \left\{ \frac{Z_{0e,o_5}\sin\theta_4}{Z_{0e,o_4}} \left( Z_{0e,o_5}Z_{2e,o}\sin2\theta_5 + 2Z_{4e,o} \right)^2 \right\} \end{matrix} \right] \tag{5d}$$

$$C_{e,o} = \frac{j}{Z_{0e,o_5}Z_{1e,o}^2}\left[\left\{\begin{array}{l}\frac{2\cos\theta_4}{Z_{0e,o_5}^2}\left(Z_{0e,o_5}Z_{2e,o}\sin2\theta_5 + 2Z_{5e,o}\right)\left(Z_{0e,o_5}Z_{2e,o}\cos2\theta_5 - (Z_{0e,o_5}^2 + Z_{3e,o})\sin2\theta_5\right)\right\} + \\ \frac{\sin\theta_4}{Z_{0e,o_4}}\left(Z_{0e,o_5}Z_{2e,o}\cos2\theta_5 - (Z_{0e,o_5}^2 + Z_{3e,o})\sin2\theta_5\right)^2\end{array}\right\} - \left\{\frac{Z_{0e,o_4}\sin\theta_4}{Z_{0e,o_5}^3}\times\left(Z_{0e,o_5}Z_{2e,o}\sin2\theta_5 + 2Z_{5e,o}\right)^2\right\}\right] \tag{5e}$$

$$D_{e,o} = \frac{1}{Z_{0e,o_5}Z_{1e,o}^2}\left[\begin{array}{l}\left\{Z_{0e,o_5}Z_{2e,o}\cos2\theta_5 - (Z_{0e,o_5}^2 + Z_{3e,o})\sin2\theta_5\right\}\left\{\begin{array}{l}\cos\theta_4\left(Z_{0e,o_5}Z_{2e,o}\cos2\theta_5 - (Z_{0e,o_5}^2 + Z_{3e,o})\sin2\theta_5\right)\\ -\frac{\sin\theta_4}{Z_{0e,o_4}}\left(Z_{0e,o_5}Z_{2e,o}\sin2\theta_5 + 2Z_{4e,o}\right)\\ -\frac{Z_{0e,o_4}\sin\theta_4}{Z_{0e,o_5}^2}\left(Z_{0e,o_5}Z_{2e,o}\sin2\theta_5 + 2Z_{5e,o}\right)\end{array}\right\}\\ -\left\{\frac{\cos\theta_4}{Z_{0e,o_5}}\left(Z_{0e,o_5}Z_{2e,o}\sin2\theta_5 + 2Z_{4e,o}\right)\left(Z_{0e,o_5}Z_{2e,o}\sin2\theta_5 + 2Z_{5e,o}\right)\right\}\end{array}\right] \tag{5f}$$

where,

$$Z_{1e,o} = Z_{0e_6}\cot\theta_6 + Z_{0o_6}\tan\theta_6, Z_{2e,o} = Z_{0e_6}\cot\theta_6 - Z_{0o_6}\tan\theta_6, Z_{3e,o} = Z_{0e,o_6}Z_{0o,e_6},$$
$$Z_{4e,o} = Z_{0e,o_6}Z_{0o,e_6}\cos^2\theta_5 - Z_{0e,o_5}^2\sin^2\theta_5, Z_{5e,o} = Z_{0e,o_5}^2\cos^2\theta_5 - Z_{0e,o_6}Z_{0o,e_6}\sin^2\theta_5 \ X$$

Moreover, the *S*-parameter expressions related to through and coupling can be found after applying the matching and isolation conditions from (4).

## 3. Coupler design methodology

A traditional 90° SSDC, as depicted in Fig 1, typically offers a loose coupling of approximately 8.5 dB, with a maximum coupling bandwidth of 60–70%. It maintains a consistent 90° phase difference between the through and coupled ports. Scattering parameters $|S_{21}|$ and $|S_{41}|$ of an ideal backward wave directional coupler are [32]:

$$\left|S_{21}\right| = \frac{\sqrt{1-k^2}}{\sqrt{1-k^2}\cos\theta + j\sin\theta}, \ \left|S_{41}\right| = \frac{jk\sin\theta}{\sqrt{1-k^2}\cos\theta + j\sin\theta} \tag{6}$$

where $\theta = \beta l$, and $k = \frac{Z_{0e}-Z_{0o}}{Z_{0e}+Z_{0o}}$. The maximum coupling occurs when

$$\theta = \beta l = \frac{\pi}{2} \ or \ l = \frac{\lambda_g}{4}$$

where $\lambda_g$ denotes the guided wavelength in the medium of the transmission line. Various strategies have been proposed to enhance the coupling of directional couplers in [12–16,28,30,31]. One standard method involves integrating an additional coupled line into one of the branches of an existing coupled-line structure, as shown in Fig 1 by SSDC-ACL, with either a similar or a different coupled-line gap or coupled-line shape [34].

In this case, two scenarios arise: either $\theta_1$ and $\theta_2$ are both 90°, or $2(\theta_1+\theta_2)$ equals 90°. When $\theta_1$ and $\theta_2$ are both 90°, coupling increases significantly, but this comes at the cost of reduced bandwidth and increased insertion loss, as depicted in Fig 6. While this approach significantly enhances coupling up to 4 dB, it reduces the coupling bandwidth to 25% and alters phase characteristics, as in hybrid couplers. Conversely, when $2(\theta_1+\theta_2)$ is 90°, for smaller values of $\theta_2$, the coupling and bandwidth remain stable, while loss and directivity improve. However, in both cases, the asymmetry of the structure results in the loss of the 90° phase difference between coupled and through ports.

Another approach is shown as SSDC-SCL in Fig 1, where an additional coupled line is added to both branches with $2(\theta_1+\theta_2)$ equals 90°, the structure retains its symmetry, allowing the 90° phase difference to be preserved, along with

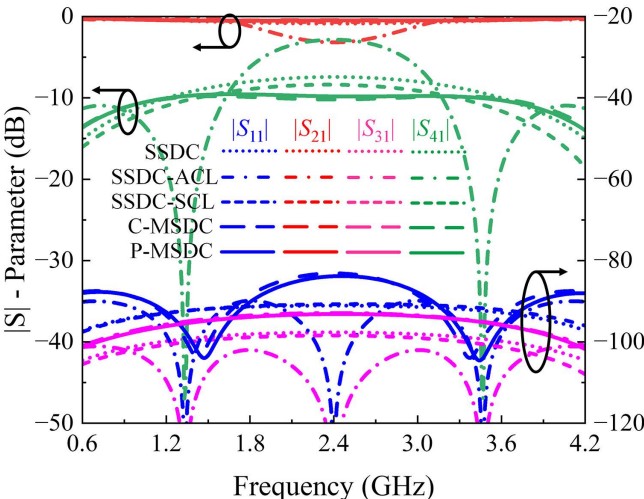

**Fig 6. Calculated|*S*| – parameter response of SSDC, SSDC – ACL, SSDC – SCL, C – MSDC and P – MSDC.**

improved loss and directivity and unchanged coupling and bandwidth performance, again for smaller values of $\theta_2$. As a result, a symmetrical coupled-line directional coupler with additional coupled lines offers a good balance, providing desired coupling, improved loss, better directivity, and the desired bandwidth performance. In Figs 6 and 7, the theoretical *S*-parameters, phase, and directivity responses of all SSDC configurations in Fig 1 are shown. Also, the variations in the coupling coefficient and insertion loss, the phase difference between the through and coupled ports, and the directivity for different values of $\theta_2$ in SSDC – ACL and SSDC – SCL are shown in Figs 8 and 9, respectively.

To achieve consistent coupling across a broader frequency range than what a single-section coupler can provide, multiple coupled sections need to be connected in succession. Each section is designed to be a quarter wavelength long at the center frequency. Through careful selection of even and odd-mode impedances for each section, the coupler's

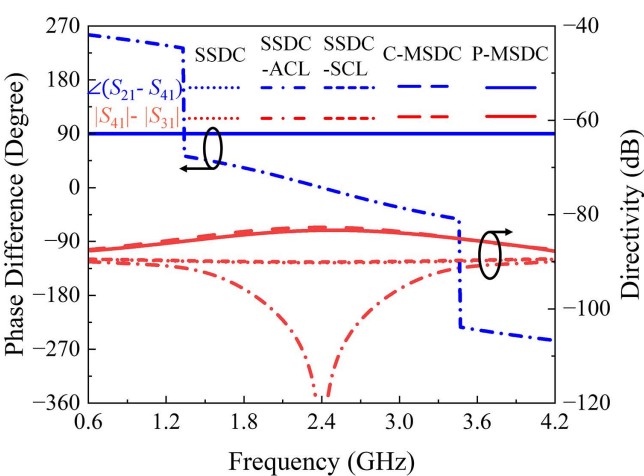

**Fig 7. Calculated phase difference and directivity response of SSDC, SSDC – ACL, SSDC – SCL, C – MSDC and P – MSDC.**

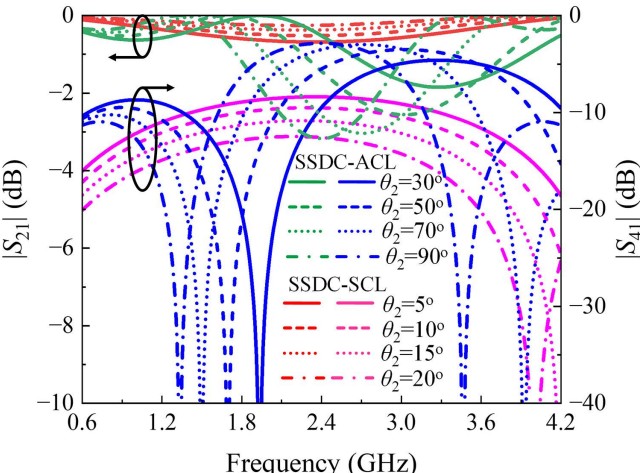

**Fig 8. Calculated coupling and insertion loss variation for different $\theta_2$ of SSDC − ACL and SSDC − SCL.**

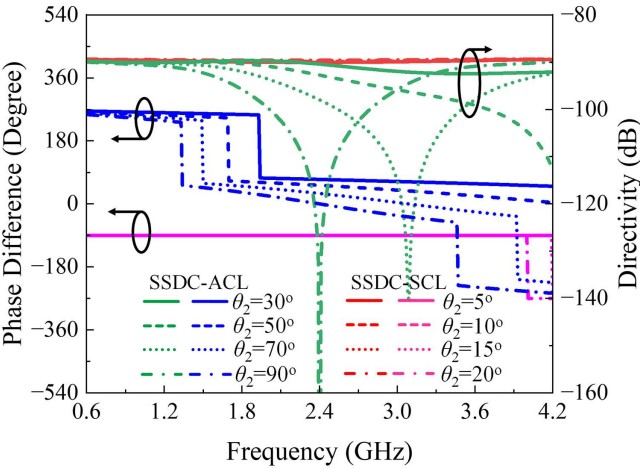

**Fig 9. Calculated phase difference and directivity variation for different $\theta_2$ of SSDC − ACL and SSDC − SCL.**

bandwidth can be expanded. This conventional multi-section symmetrical directional coupler (C-MSDC) approach is one of the most effective solutions for maintaining a wideband response with 90° phase difference, but at the cost of loose coupling, as shown in Fig 6. Table 2 presents the even- and odd-mode impedances of a traditional three-section symmetrical directional coupler with a coupling level of 10 dB and ripple of ± 0.2 dB, while maintaining the manufacturing limitation of 0.10 mm. This design values allow us to create a small wideband directional coupler.

Insertion loss function or power loss ratio $L$ of coupler [32] is an important parameter to understand the wideband behavior and is expressed as

$$L = \frac{1}{|S_{21}|^2}$$

(7)

**Table 2. Design Parameters for 0.2 dB Ripple 10 dB Symmetrical Coupler with Three Sections.**

| $Z_{0e5}$ | $Z_{0e6}$ | $Z_{0o5}$ | $Z_{0o6}$ | $\theta_4$ | $\theta_5$ | $\theta_6$ |
|---|---|---|---|---|---|---|
| 53.47 | 78.71 | 46.75 | 31.76 | 90 | 30 | 15 |

('Z' in Ω & 'θ' in degrees.)

where $|S_{21}|$ is the scattering parameter between ports 1 and 2 of the equivalent four-port directional coupler. For an ideal directional coupler

$$\left|S_{11}\right| = \left|S_{31}\right| = 0 \tag{8}$$

which implies

$$|S_{21}|^2 + |S_{41}|^2 = 1 \tag{9a}$$

and

$$L = \frac{1}{1 - |S_{41}|^2} \tag{9b}$$

In an ideal directional coupler, it is desired that $|S_{21}|$ and $|S_{31}|$ remain constant across a specified frequency range. This requires function L to be constant over this frequency band and take any form outside this frequency range.

To achieve a tight coupling while preserving the wideband behavior and actual size of the overall circuit, a modified coupler structure featuring circular-shaped coupled line structures in all four outer sections of a three-section multi-section coupler has been proposed (P-MSDC). The length of the additional coupled-line structure is kept as small as possible for unchanged coupling bandwidth, as discussed earlier, and the circular shape is chosen to avoid discontinuities while keeping the overall size of the coupler structure smaller.

The primary design objective is to improve coupling while maintaining bandwidth and a 90° phase difference with very good directivity, which are critical performance parameters of a directional coupler. Additionally, it is essential to ensure good performance in terms of return loss and isolation. Initial dimensions are chosen from [8]. The closed-form equations are applied to determine the design parameters of the proposed coupler. Subsequently, the overall performance is fine-tuned through 3-D electromagnetic analysis with a focus on minimizing the lengths of additional coupled line structures by slightly adjusting the preliminary section geometry. This adjustment aims to maximize coupling and maintain a 90° phase difference while ensuring ultra-wideband behavior, size reduction, improved return loss, and directivity response. The electrical length reduction percentage is calculated for the proposed work and previously reported works as shown in Table 3. Compared to [15,16,18,26,27,29], the proposed multi-section directional coupler exhibits a size reduction of 22%.

Having calculated S-parameters, phase differences, and directivity responses for all structures, it is evident that although the coupling value is smaller in P–MSDC, it offers maximum coupling bandwidth with minimal ripples and minimal insertion loss amongst all topologies. Moreover, the return loss, isolation, and directivity performance are better than C-MSDC. A constant 90° phase difference is offered by all structures, except SSDC-ACL, as expected due to symmetrical topologies. Various response specifications of SSDC, SSDC-ACL, SSDC-SCL, C-MSDC, and P-MSDC are also listed in Table 4, where indices show that in P-MSDC, 1.2 dB improvement in coupling at the cost of 4% reduction in bandwidth than C-MSDC, as expected. The proposed structure has been fabricated and tested for various coupling and ripple values, demonstrating improved coupling without significantly affecting the coupler's ultra-wideband performance, as indicated by the simulated S-parameters, phase, and directivity responses shown in Figs 10 and 11.

**Table 3. Comparative analysis of various topologies of directional couplers.**

| Ref. | Design parameters | Total series length | Reduction % |
|---|---|---|---|
| [15] | $Z_{0e1}=56.50\ \Omega$, $Z_{0o1}=44.25\ \Omega$, $Z_{0e2}=81.85\ \Omega$, $Z_{0o2}=30.54\ \Omega$, $Z_{0e3}=56.50\ \Omega$, $Z_{0o3}=44.25\ \Omega$, $\theta_1=90°$, $\theta_2=90°$, $\theta_3=90°$ | 270° | No reduction |
| [16] | $Z_{0e1}=58.8\ \Omega$, $Z_{0o1}=42.5\ \Omega$, $Z_{0e2}=78.5\ \Omega$, $Z_{0o2}=31.8\ \Omega$, $Z_{0e3}=233\ \Omega$, $Z_{0o3}=10.7\ \Omega$, $Z_{0e4}=78.5\ \Omega$, $Z_{0o4}=31.8\ \Omega$, $Z_{0e5}=58.8\ \Omega$, $Z_{0o5}=42.5\ \Omega$, $\theta_1=\theta_2=\theta_3=\theta_4=\theta_5=90°$ | 450° | No reduction |
| [18] | $Z_{0e1}=54.8\ \Omega$, $Z_{0o1}=45.6\ \Omega$, $Z_{0e2}=209\ \Omega$, $Z_{0o2}=12.0\ \Omega$, $Z_{0e3}=54.8\ \Omega$, $Z_{0o3}=45.6\ \Omega$, $\theta_1=90°$, $\theta_2=90°$, $\theta_3=90°$ | 270° | No reduction |
| [26] | $Z_{0e1}=61.5\ \Omega$, $Z_{0o1}=40.7\ \Omega$, $Z_{0e2}=174.9\ \Omega$, $Z_{0o2}=14.3\ \Omega$, $Z_{0e3}=61.5\ \Omega$, $Z_{0o3}=40.7\ \Omega$, $\theta_1=90°$, $\theta_2=90°$, $\theta_3=90°$ | 270° | No reduction |
| [27] | $Z_{0e1}=87.71\ \Omega$, $Z_{0o1}=28.50\ \Omega$, $Z_{0e2}=74.10\ \Omega$, $Z_{0o2}=33.74\ \Omega$, $Z_{0e3}=64.79\ \Omega$, $Z_{0o3}=38.59\ \Omega$, $Z_{0e4}=58.52\ \Omega$, $Z_{0o4}=42.72\ \Omega$, $Z_{0e5}=54.43\ \Omega$, $Z_{0o5}=45.92\ \Omega$, $Z_{0e6}=51.91\ \Omega$, $Z_{0o6}=48.15\ \Omega$, $\theta_1=\theta_2=\theta_3=\theta_4=\theta_5=\theta_6=90°$ | 540° | No reduction |
| [29] | $Z_{0e1}=67.14\ \Omega$, $Z_{0o1}=37.24\ \Omega$, $Z_{0e2}=59.86\ \Omega$, $Z_{0o2}=41.76\ \Omega$, $Z_{0e3}=55.06\ \Omega$, $Z_{0o3}=45.41\ \Omega$, $Z_{0e4}=52.11\ \Omega$, $Z_{0o4}=47.97\ \Omega$, $\theta_1=\theta_2=\theta_3=\theta_4=90°$ | 360° | No reduction |
| **This work** | $Z_0e_4=78.71\ \Omega$, $Z_0o_4=31.76\ \Omega$, $Z_0e_5=53.47\ \Omega$, $Z_0o_5=46.75\ \Omega$, $Z_0e_6=78.71\ \Omega$, $Z_0o_6=31.76\ \Omega$, $\theta_4=90°$, $\theta_5=30°$, $\theta_6=15°$, | **210°** | 22% reduction |

**Table 4. Specifications of different directional couplers.**

| Structure type | C.±R. (dB) | BW (%) | Iso. (dB) | R.L. (dB) | P.D.±P.I. (deg.) | Dir. (dB) |
|---|---|---|---|---|---|---|
| SSDC* | 8±0.5 | 67 | >97 | >90 | 90° | >89 |
| SSDC-ACL* | 3±0.5 | 25 | >96 | >90 | 0° @ $f_0$ | >88 |
| SSDC-SCL* | 9±0.5 | 65 | >98 | >90 | 90° | >89 |
| C-MSDC* | 10±0.2 | 103 | >92 | >82 | 90° | >82 |
| C-MSDC# | 9.2±0.8 | 122 | >16 | >10 | 90±8° | >20 |
| P-MSDC* | 9.8±0.2 | 102 | >93 | >83 | 90° | >83 |
| P-MSDC# | 8±0.8 | 118 | >15 | >10 | 90±5° | >26 |

C.±R.-Coupling±Ripple, B.W.-Bandwidth, Iso. -Isolation, R.L.-Return Loss, P.D.-Phase Difference, P.I.-Phase Imbalance, Dir.-Directivity, *-Calculated Response, #-Simulated Response.

To strengthen the effectiveness of proposed coupler unit, various other key performance indicators such as losses, impedance matching, group delay variation and sensitivity etc., are analysed. Along with the excellent insertion loss of 0.45–1.62 dB, analyzing the separate loss components offers a better understanding of the coupler's functioning. As a result, conductor, dielectric, and radiation losses were investigated using field-based loss analysis in HFSS, as seen in Fig 12. Looking at the figure, it is clear that the radiation loss total loss is much lower than 0.15 and 0.21 throughout the whole UWB band.

Sensitivity analysis with ±10% variation in the coupling gap has been carried out. As expected, the coupling level is the most sensitive parameter and shows noticeable variation, which is illustrated in the corresponding plots, shown in Fig 13. Other performance metrics, such as return loss, isolation, and insertion loss, are comparatively less affected, demonstrating the robustness of the design.

The simulated Smith chart representation, shown in Fig 14, confirms acceptable impedance matching over the entire operating bandwidth from 0.75 to 3.75 GHz. Moreover, regarding signal integrity, the simulated group delay response is presented in Fig 15, where the variation for the coupled port is observed to be less than 1 ns across the operating band, which is well within 0.4 to 0.7 ns, indicating good phase linearity and suitability for UWB applications.

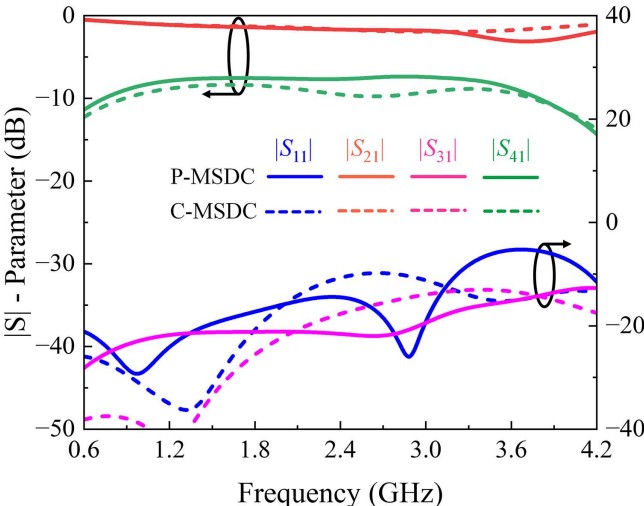

**Fig 10. Full-wave simulated |S| – parameter response of conventional and proposed multi-section directional couplers.**

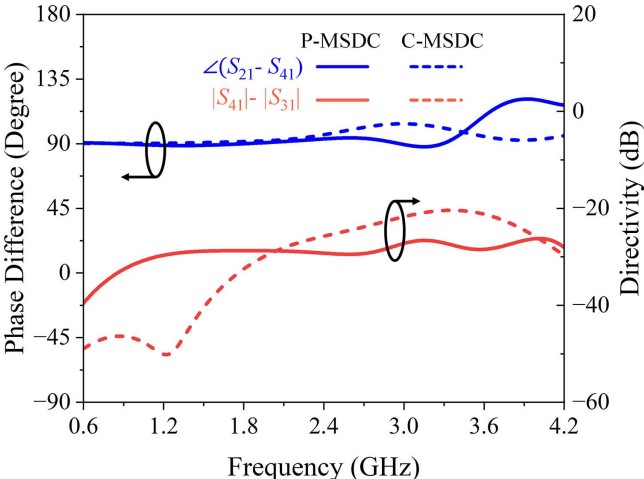

**Fig 11. Full-wave simulated phase difference and directivity response of conventional and proposed multi-section directional couplers.**

## 4. Experimental results and discussion

To assess the performance of the three-section symmetric coupler for higher coupling and ultra-wideband applications, a prototype is fabricated on a 60 mil-thick Rogers RO4003C substrate with a dielectric constant of 3.55 and a loss tangent of 0.0027 [36] operating at a center frequency of $f_0 = 2.4$ GHz. The Layout view and a photograph of the fabricated three-section directional coupler, along with physical dimensions, are shown in Figs 16 and 17, respectively.

The HFSS full-wave simulator is used to obtain the final physical dimensions of the coupler. Measured and simulated *S*-parameters, phase difference, and directivity response of the presented three-section directional coupler are shown in Figs 18 and 19. Also, the key response parameters of P-MSDC are listed in Table 5. The results corroborate good performance of the prototype in the 0.75 GHz to 3.75 GHz band. There is a good match between simulated and measured

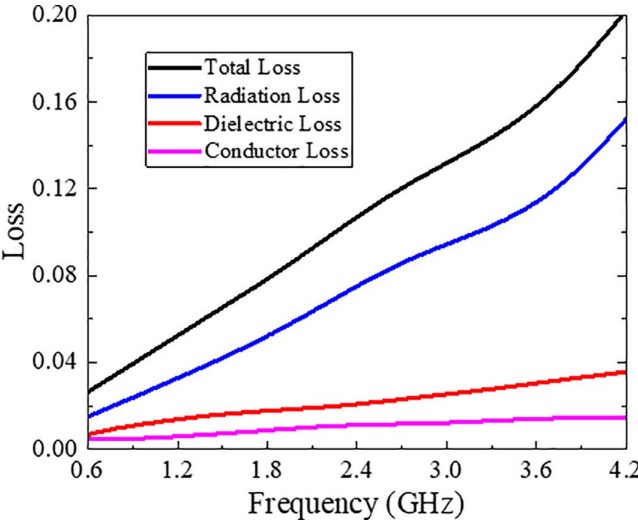

**Fig 12. Losses of proposed multi-section directional coupler.**

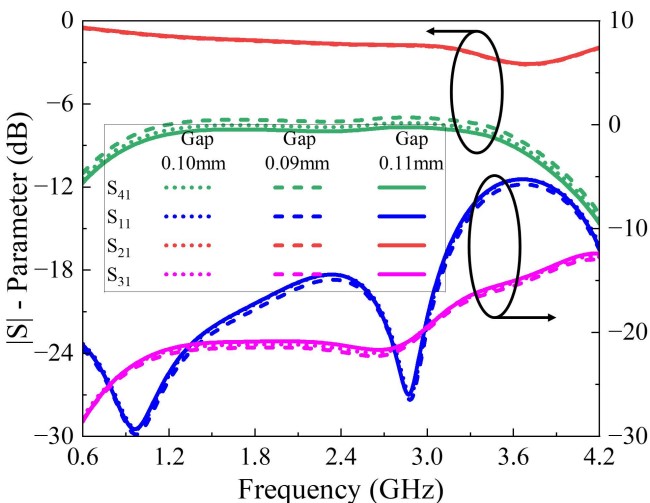

**Fig 13. Sensitivity analysis of proposed multi-section directional coupler.**

results. Focusing on the critical parameters like coupling ± ripple, ultra-wideband performance, return loss, directivity, isolation, and phase difference, it is observed that coupling is 7.8 dB, 8.1 dB, and 8.3 dB with 0.7 dB, 1 dB, and 1.4 dB ripples, respectively, and the effective operational bandwidth achieved is 104%, 123% and 133%. The phase difference is 90° across the entire band except at the end, where it deviates from 90° by no more than ±4°. The measured insertion loss in the desired range varies from 0.45 to 1.62 dB, reflecting the constant nature of the power-loss function $L$ of the fabricated coupler, as desired for any ultra-wideband coupler, discussed in (5).

Major performance indicators for the previous related works and our results are tabulated in Table 6. It is evident that our design significantly outperforms the designs [15,16,18,28] in terms of geometry and fabrication complexity, [13,17,22,26,28,29] in terms of coupling ± ripple and/or bandwidth performance, [16,22] in phase response. Measured

**Fig 14. Impedance matching of proposed multi-section directional coupler.**

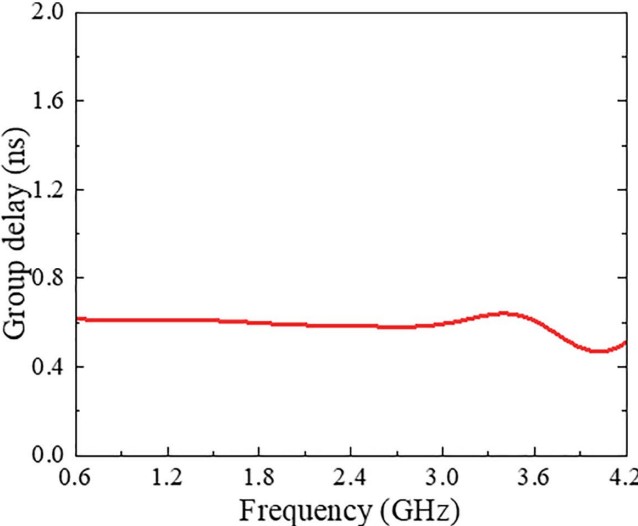

**Fig 15. Group delay variation of proposed multi-section directional coupler.**

directivity exceeds 30 dB throughout the ultra-wideband, which is one of the primary performance indicators for coupling structures. Although few designs appear to surpass ours in terms of isolation and return losses, this is due to the additional discontinuities introduced by the four extra coupled line sections in the proposed structure. Nevertheless, these parasitic effects can be further addressed by the parasitic compensation method as in [24–27]. However, all the studies listed in Table 6 used a non-classical directional coupler structure, except for the presented work.

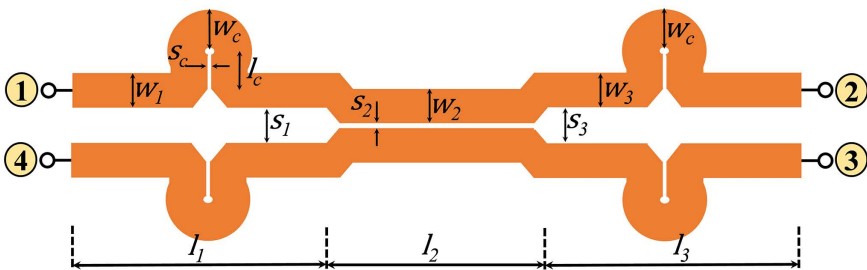

**Fig 16. Layout of the proposed multi-section directional coupler.**

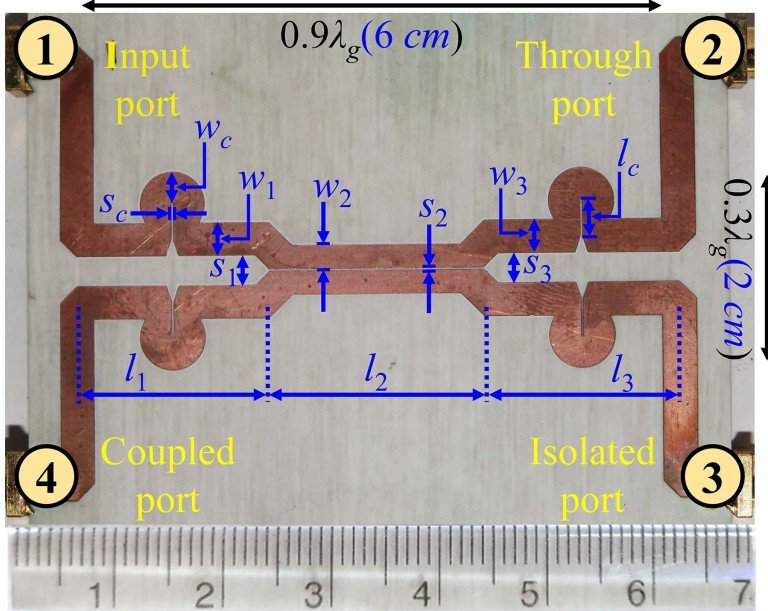

**Fig 17. Photograph of the manufactured prototype of the suggested multi-section directional coupler.** Optimized dimension: $s_c$=0.1, $w_c$=3.36, $l_c$=3.3, $s_1$=$s_3$=2.86, $w_1$=$w_3$=3.36, $s_2$=0.1, $w_2$=2.41, $l_1$=$l_3$=19.2, $l_2$=21.6.; unit: mm.

## 5. Conclusion

A circular-shaped coupled line structure based on a tightly coupled ultra-wideband symmetric three-section directional coupler topology with all inherent properties of a classical directional coupler is presented here to operate within the ultra-wideband spectrum range of 0.75 to 3.75 GHz with a bandwidth of 5:1 and coupling of 8.3 dB with ± 1.4 dB ripple. Using a conventional microstrip line-based PCB manufacturing technique, this structure exhibits good coupling, phase difference, and bandwidth characteristics across the entire band with sufficient return loss and directivity. Consequently, our proposed circuit not only surpasses the fabrication complexity and complex design process of similar designs but also achieves bandwidths of 3.18:1, 4.2:1, and 5:1, which exceed the theoretical limit of 3.13:1 [8], and a suitable candidate to complement hybrid UWB front-end systems [37]. The rectangular occupied area of the coupler unit is $0.067\lambda_0^2$ at the mid-band frequency.

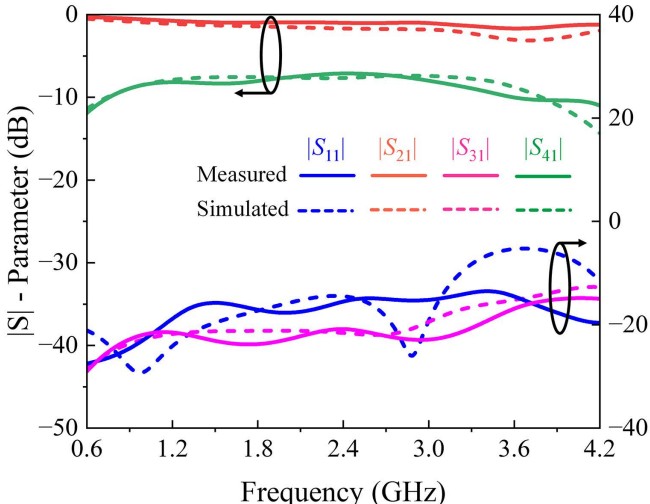

**Fig 18. Full-wave simulated and measured |S| – parameter response of the suggested multi-section directional coupler.**

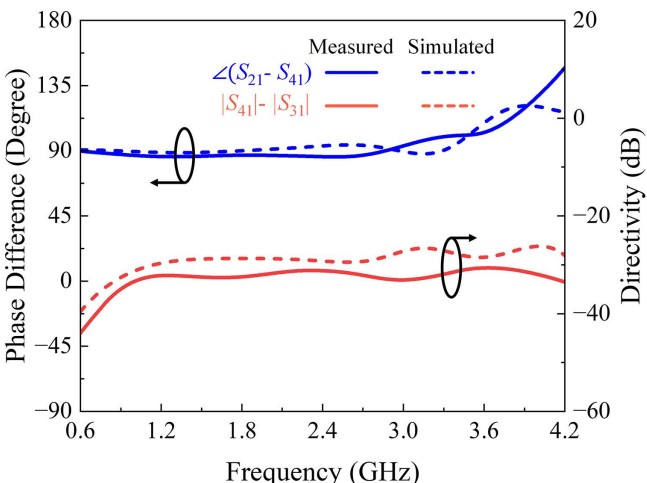

**Fig 19. Full-wave simulated and measured phase difference and directivity response of the suggested multi-section directional coupler.**

**Table 5. Measured response parameters of proposed coupler.**

| Structure type | C.±R. (dB) | B.W. (%) | Iso. (dB) | R.L. (dB) | P.D.±P.I. (deg.) | Dir. (dB) |
|---|---|---|---|---|---|---|
| P-MSDC | 7.8±0.7<br>8.1±1<br>8.3±1.4 | 105<br>123<br>133 | >20 | >16 | 90±4 | >30 |

**Table 6. Comparison of the proposed coupler with previous UWB coupler works.**

| Ref. | Techniques | C.±R. (dB) | Freq. range (GHz) | B.W. (%) | Iso. (dB) | R.L. (dB) | P.D. (deg.) | Size ($\lambda_0^2$) |
|------|-----------|-----------|------------------|----------|-----------|-----------|-------------|-----------|
| [13] | Slot coupled | 6.4±0.4 | 1.2-1.8 | 40 | 30 | 25 | N.R. | 0.67 |
| [15] | Tandem coupler | 3±0.3 | 0.45 - 2.2 | 132 | 15 | 15 | N.R. | N.R. |
| [16] | VIP coupler | 3±0.65 | 1 - 9 | 160 | 20 | 20 | 90±5° | N.R. |
| [17] | Slot coupled | 3±1<br>6±1.4<br>10±1.5 | 3.1 - 10.6 | 109 | 23<br>20<br>19 | 21<br>18<br>19 | N.R. | 0.19 |
| [18] | Slot coupled | 3±1 | 2.3 - 12.3 | 140 | 23 | 23 | 90±2.5° | 0.89 |
| [22] | Non-uniform Coupled Line | 3±0.7 | 4.5 - 17.4 | 100 | 14 | 14 | 90±6° | N.R. |
| [26] | Parasitic compensation | 3±0.25 | 1.15 - 5.1 | 79 | 29 | 26 | 90±2° | N.R. |
| [27] | Parasitic compensation | 10±0.8 | 2 - 22 | 166 | 18 | 19 | 90±2° | N.R. |
| [28] | Defected Ground Structure | 3±N.R. | 0.45 - 4.5 | 160 | 28 | 28 | N.R. | N.R. |
| [29] | Dielectric overlay | 15±1 | 15 - 45 | 100 | 10 | 10 | N.R. | N.R. |
| **This work** | **Coupled Microstrip Line** | **7.8±0.7<br>8.1±1<br>8.3±1.4** | **0.75–3.75** | **105<br>123<br>133** | **20** | **16** | **90±4°** | **0.067** |

N.R. – Not reported.

## Author contributions

**Conceptualization:** Shrawan K. Patel, Rusan Kumar Barik, Niraj Kumar Dewangan, Slawomir Koziel.

**Data curation:** Shrawan K. Patel.

**Formal analysis:** Shrawan K. Patel, Rusan Kumar Barik, Niraj Kumar Dewangan, Slawomir Koziel.

**Methodology:** Shrawan K. Patel, Rusan Kumar Barik, Niraj Kumar Dewangan, Slawomir Koziel.

**Software:** Shrawan K. Patel, Rusan Kumar Barik, Niraj Kumar Dewangan.

**Supervision:** Shrawan K. Patel, Rusan Kumar Barik, Niraj Kumar Dewangan, Slawomir Koziel.

**Validation:** Shrawan K. Patel, Rusan Kumar Barik, Niraj Kumar Dewangan, Slawomir Koziel.

**Visualization:** Shrawan K. Patel, Rusan Kumar Barik, Niraj Kumar Dewangan, Slawomir Koziel.

**Writing – original draft:** Shrawan K. Patel, Rusan Kumar Barik.

**Writing – review & editing:** Rusan Kumar Barik, Niraj Kumar Dewangan, Slawomir Koziel.

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
