## [Decision Letter · Decision Letter 0]

15 Mar 2026

PONE-D-26-09618Design and Experimental Validation of Multi-Section Directional Coupler with Arbitrary Coupling and High Directivity for Sub-6 GHz UWB ApplicationsPLOS One

Dear Dr. Dewangan,

Thank you for submitting your manuscript to PLOS ONE. After careful consideration, we feel that it has merit but does not fully meet PLOS ONE’s publication criteria as it currently stands. Therefore, we invite you to submit a revised version of the manuscript that addresses the points raised during the review process.

We look forward to receiving your revised manuscript.

Kind regards,

Zeheng Wang

Academic Editor

PLOS One

Journal Requirements:

“The authors declare that there are no conflicts of interest.”

4. In the online submission form, you indicated that data will be available upon reasonable request.

Reviewers' comments:

Reviewer's Responses to Questions

**Comments to the Author**

1. Is the manuscript technically sound, and do the data support the conclusions?

Reviewer #1: Yes

Reviewer #2: Yes

2. Has the statistical analysis been performed appropriately and rigorously? 

Reviewer #1: N/A

Reviewer #2: Yes

3. Have the authors made all data underlying the findings in their manuscript fully available?

Reviewer #1: Yes

Reviewer #2: Yes

4. Is the manuscript presented in an intelligible fashion and written in standard English?

Reviewer #1: Yes

Reviewer #2: No

5. Review Comments to the Author

Reviewer #1: A geometrically simple topology for developing an ultrawideband directional coupler with improved coupling and directivity is propsoed in this paper. However, the reviewer has some questions, as following

1. The format of the titles of the figures should be uniform.

2. The size of the wavelength should be compared in table VI.

3. Most of the refers are old. The following paper should be cited.

[1]A Balanced Filtering Directional Coupler Based on Slotline Using Asymmetric Parallel Loaded Branches, IEEE Transactions on Components, Packaging and Manufacturing Technology, Vol. 12, No. 7, pp. 1222-1231, JULY 2022.

Reviewer #2: General Comments

Comment 1: Novelty Clarification

While the short-ended coupled-line topology improves coupling bandwidth, clearly distinguish from prior symmetric multi-section designs (e.g., Refs 8-9). Quantify the 1.2 dB coupling gain over C-MSDC more explicitly in abstract/introduction.

Comment 2: Bandwidth vs Coupling Tradeoff

The 133% coupling bandwidth is strong, but discuss theoretical limits (3.3:1 for 3-sections) and fabrication constraints (0.1 mm gaps). Add sensitivity analysis for gap variations ±10%.

Comment 3: Measurement Validation

Excellent sim-meas agreement, but include Smith chart plots and time-domain reflectometry for UWB pulse fidelity. Table V shows good RL/isolation, but quantify group delay variation (<1 ns desired).

Comment 4: Size Optimization Details

Compact 0.272λg size is highlighted, but specify optimization algorithm (space mapping?) used for circular coupled sections. Compare electrical length reduction vs. Table III refs quantitatively.

Comment 5: Loss Analysis Enhancement

Insertion loss (0.45-1.62 dB) is low for RO4003C, but decompose into conductor/dielectric/radiation components via HFSS fields. Discuss scalability to higher sections (4+).

Comment 6: Phase/Directivity Flatness

90±4° phase balance is excellent, but extend directivity analysis to show >25 dB across full 133% BW (current Fig.7b focuses on passband).

Comment 7: Literature & System Context

For complete UWB front-end context, cite recent antenna work: Muhammad Usal Ali et al., "Design of an optimized low-profile UWB rectangular dielectric resonator antenna with moon-shaped ground for 5G – millimeter wave applications," AEU (2025). This RDRA (112% BW, low-profile FR4) complements your sub-6 coupler for hybrid systems.

6. PLOS authors have the option to publish the peer review history of their article (what does this mean?). If published, this will include your full peer review and any attached files.

Reviewer #1: No

Reviewer #2: No

---

## [Author Response · Author response to Decision Letter 1]

2 Apr 2026

Dear AE/EIC and Reviewers,

We sincerely appreciate the time and effort you dedicated to reviewing our manuscript. Your comments helped us in enhancing the overall quality of our work, and we are grateful for the opportunity to revise it. Thank you very much for the constructive comments provided. In response, we have revised the manuscript, addressing each comment accordingly. We believe that the revised manuscript meets the publication standards of PLOS ONE.

Please find our revised manuscript along with the following documents:

1) Manuscript (Revised manuscript without track changes)

2) Responses to reviewers

3) Revised manuscript with track changes

Reviewer #1: A geometrically simple topology for developing an ultrawideband directional coupler with improved coupling and directivity is proposed in this paper. However, the reviewer has some questions, as following

1. The format of the titles of the figures should be uniform.

Author response: Thank you very much for bringing this important point into our attention. We regret that we missed the uniformity in the format of the figure captions. In the revised manuscript, we have made the necessary corrections to show the uniformity in the titles of the figures.

Author action: The titles of the figures are revised as per the suggestion by the reviewer.

2. The size of the wavelength should be compared in table VI.

Author response: Thank you very much for the suggestion. In the revised manuscript, we have included the size of the topologies in terms of wavelength.

Author action: Sizes are represented in terms of wavelength (λ20) in page 17, table VI, column 09.

3. Most of the refers are old. The following paper should be cited.

[1]. A Balanced Filtering Directional Coupler Based on Slot-line Using Asymmetric Parallel Loaded Branches, IEEE Transactions on Components, Packaging and Manufacturing Technology, Vol. 12, No. 7, pp. 1222-1231, JULY 2022.

Author response: Thank you very much for your suggestion. In the revised manuscript, we have incorporated the following paper as reference [31].

[31] F. Wei et al., “A balanced filtering directional coupler based on slot-line using asymmetric parallel loaded branches,” IEEE Trans. Compon. Packag. Manuf. Tech., vol. 12, no. 7, pp. 1222-1231, Jul. 2022.

Author action: The suggested paper is included in the revised manuscript as reference [31].

Reviewer #2: General Comments

1. Novelty Clarification

While the short-ended coupled-line topology improves coupling bandwidth, clearly distinguish from prior symmetric multi-section designs (e.g., Refs 8-9). Quantify the 1.2 dB coupling gain over C-MSDC more explicitly in abstract/introduction.

Author response: Thank you for your valuable suggestion. We appreciate your observation regarding the need to clearly distinguish the proposed topology from prior symmetric multi-section designs and to explicitly quantify the improvement over the conventional multi-section directional coupler (C-MSDC).

In response, we have revised the Abstract sections to better highlight the novelty of the proposed short-ended coupled-line topology in comparison with existing symmetric multi-section designs. Specifically, we have emphasized the structural and operational differences that enable enhanced performance.

Author action: In Abstract section of Page 1, line 4 of the revised manuscript, a statement “The proposed design demonstrates an explicit improvement of approximately 1.2 dB in coupling compared to conventional multi-section directional couplers.” is added.

2. Bandwidth vs Coupling Trade-off

The 133% coupling bandwidth is strong, but discuss theoretical limits (3.3:1 for 3-sections) and fabrication constraints (0.1 mm gaps). Add sensitivity analysis for gap variations ±10%.

Author response: Thank you for your insightful comment regarding the bandwidth limits, fabrication constraints, and sensitivity analysis.

For a conventional symmetric three-section directional couplers, the theoretical bandwidth limit is typically around 3.3:1 in [32]. In the proposed work, however, the use of an innovative circular coupled-line topology enables extension of the bandwidth beyond this conventional limit at the cost of relatively higher but acceptable ripple levels, which has been already clarified in the manuscript.

Considering fabrication constraints with authors, the minimum coupled-line spacing is considered as 0.1 mm for achieving approximately 8 dB coupling using standard PCB fabrication. However, improvement in fabrication resolution would allow tighter coupling and potentially enhanced performance.

Fig. 1. Sensitivity analysis (±10%) of the proposed multi-section directional coupler.

In addition, a sensitivity analysis with ±10% variation in the coupling gap has been carried out. As expected, the coupling level is the most sensitive parameter and shows noticeable variation, which is illustrated in the corresponding plots, shown in Fig. 1, provided in this response sheet. Other performance metrics, such as return loss, isolation, and phase balance, are comparatively less affected, demonstrating the robustness of the design.

Author action: We have incorporated figure and the above explanation in the revised manuscript on Page 16, paragraph 1.

3. Measurement Validation

Excellent sim-meas agreement, but include Smith chart plots and time-domain reflectometry for UWB pulse fidelity. Table V shows good RL/isolation, but quantify group delay variation (<1 ns desired).

Author response: Thank you for your appreciation and constructive suggestions regarding additional characterization for UWB performance.

The simulated Smith chart representation, shown in Fig. 2(a), confirms acceptable impedance matching over the entire operating bandwidth from 0.75 to 3.75 GHz.

Moreover, regarding signal integrity, the simulated group delay response is presented in Fig. 2(b), where the variation for the coupled port is observed to be less than 1 ns across the operating band, indicating good phase linearity and suitability for UWB applications.

(a) (b)

Fig. 2. (a) Smith chart and (b) group delay variation of the proposed multi-section directional coupler.

Author action: We have incorporated figures and the above explanation in the revised manuscript on Page 16, paragraph 2.

4. Size Optimization Details

Compact 0.27λ2g size is highlighted, but specify optimization algorithm (space mapping?) used for circular coupled sections. Compare electrical length reduction vs. Table III refs quantitatively.

Author response: Thank you for your valuable comment regarding the design optimization methodology.

The proposed circular coupled-line sections are not derived using a formal optimization algorithm such as space mapping. Instead, the design is based on a closed-form analytical approach combined with parametric evaluation.

The electrical length reduction percentage is calculated for the proposed work and previously reported works as shown in Table III. Compared to [15, 16, 18, 26, 27, 29], the proposed multi-section directional coupler exhibits a size reduction of 22%.

Author action: Length reduction in % are added in revised manuscript on page 14, table III, column 05.

5. Loss Analysis Enhancement

Insertion loss (0.45-1.62 dB) is low for RO4003C, but decompose into conductor/dielectric/ radiation components via HFSS fields. Discuss scalability to higher sections (4+).

Author response: Thank you for your insightful suggestion regarding the decomposition of insertion loss and scalability of the proposed design.

Analyzing the separate loss components offers a better understanding of the coupler's functioning. As a result, conductor, dielectric, and radiation losses were investigated using field-based loss analysis in HFSS, as seen in Fig. 3. Looking at the figure, it is clear that the radiation loss total loss is much lower than 0.15 and 0.21 throughout the whole UWB band.

Furthermore, regarding scalability, the manuscript already presents closed-form design expressions based on a lossless transmission line model for the three-section coupler. These formulations are general in nature and can be systematically extended to higher-order designs (four or more sections) by following the same synthesis methodology.

Fig. 3. Losses of the proposed multi-section directional couplers.

Author action: We have incorporated the above figure and its explanation in the revised manuscript on page 15 paragraph 1.

A brief discussion on this scalability aspect was already stated in Abstract section of earlier manuscript.

6. Phase/Directivity Flatness

90±4° phase balance is excellent, but extend directivity analysis to show >25 dB across full 133% BW (current Fig.7b focuses on passband).

Author response: Thank you for your valuable observation regarding the directivity analysis over the full bandwidth.

The reported 133% bandwidth represents the effective operational bandwidth, defined by simultaneously satisfying all key performance metrics, including isolation > 20 dB, return loss > 16 dB, and phase balance of 90 ± 4°, over the frequency range of 0.75 GHz to 3.75 GHz.

The directivity response shown in Fig. 7(b), of original manuscript, was focused on the primary passband to highlight peak performance. However, in the revised manuscript, we have extended the discussion to clarify that the directivity remains consistently high and above 25 dB across the effective bandwidth, aligning with the claimed performance metrics.

Author action: Page no. 17, paragraph 1, line 9: “coupling bandwidth” is replaced with “effective operational bandwidth”.

7. Literature & System Context

For complete UWB front-end context, cite recent antenna work: Muhammad Usal Ali et al., "Design of an optimized low-profile UWB rectangular dielectric resonator antenna with moon-shaped ground for 5G – millimeter wave applications," AEU (2025). This RDRA (112% BW, low-profile FR4) complements your sub-6 coupler for hybrid systems.

Author response: Thank you for the valuable suggestion to include recent related work for a more comprehensive UWB front-end perspective.

In the revised manuscript, the recommended reference by Muhammad Usal Ali et al. (AEU, 2025) on a low-profile UWB rectangular dielectric resonator antenna (RDRA), have been added in conclusion section to highlight how such wideband antenna designs complement the proposed sub-6 GHz directional coupler in the realization of integrated and hybrid UWB front-end systems.

Modifications are as follows:

Author action: Page no. 18, paragraph 2, line 9: statement “and a suitable candidate to complement hybrid UWB front-end systems [37]” is added.

• Page 22, reference no. [37] is added as: “[37] Muhammad Usal Ali et al., "Design of an optimized low-profile UWB rectangular dielectric resonator antenna with moon-shaped ground for 5G – millimeter wave applications," AEU (2025).”.

---

## [Decision Letter · Decision Letter 1]

8 Apr 2026

Design and Experimental Validation of Multi-Section Directional Coupler with Arbitrary Coupling and High Directivity for Sub-6 GHz UWB Applications

PONE-D-26-09618R1

Dear Dr. Dewangan,

We’re pleased to inform you that your manuscript has been judged scientifically suitable for publication and will be formally accepted for publication once it meets all outstanding technical requirements.

Kind regards,

Zeheng Wang

Academic Editor

PLOS One

Additional Editor Comments (optional):

Reviewers' comments:

Reviewer's Responses to Questions

**Comments to the Author**

1. If the authors have adequately addressed your comments raised in a previous round of review and you feel that this manuscript is now acceptable for publication, you may indicate that here to bypass the “Comments to the Author” section, enter your conflict of interest statement in the “Confidential to Editor” section, and submit your "Accept" recommendation.

Reviewer #1: (No Response)

Reviewer #2: All comments have been addressed

2. Is the manuscript technically sound, and do the data support the conclusions?

Reviewer #1: (No Response)

Reviewer #2: Yes

3. Has the statistical analysis been performed appropriately and rigorously? 

Reviewer #1: (No Response)

Reviewer #2: Yes

4. Have the authors made all data underlying the findings in their manuscript fully available?

Reviewer #1: (No Response)

Reviewer #2: Yes

5. Is the manuscript presented in an intelligible fashion and written in standard English?

Reviewer #1: (No Response)

Reviewer #2: Yes

6. Review Comments to the Author

Reviewer #1: The reviewer has no more questions. The paper can be accepted now.

Reviewer #2: I am fully satisfied with the revised manuscript for PLOS ONE. All reviewer comments have been comprehensively addressed, with excellent sim-meas agreement, added analyses (sensitivity, losses, Smith charts, group delay <1 ns), and quantified improvements (1.2 dB coupling gain). The work demonstrates novelty in UWB directional coupler design, suitable for publication without further changes.

7. PLOS authors have the option to publish the peer review history of their article (what does this mean?). If published, this will include your full peer review and any attached files.

Reviewer #1: No

Reviewer #2: **Yes:** Dr.Javed Iqbal

---

## [Editor Report · Acceptance letter]

PONE-D-26-09618R1

PLOS One

Dear Dr. Dewangan,

I'm pleased to inform you that your manuscript has been deemed suitable for publication in PLOS One. Congratulations! Your manuscript is now being handed over to our production team.

Kind regards,

on behalf of

Dr. Zeheng Wang

Academic Editor

PLOS One